# A comparative study of postnatal anthropometric growth in very preterm infants and intrauterine growth

Fu-Sheng Chou [1,2] ✉, Hung-Wen Yeh [3,4] & Reese H. Clark [5]

Most growth references for very preterm infants were developed using measurements taken at birth, and were thought to represent intrauterine growth. However, it remains unclear whether the goal of approximating an intrauterine growth rate as stated by the American Academy of Pediatrics is attainable by very preterm infants. Using real-world measurement data from very preterm infants born between 2010 through 2020, we develop models to characterize the patterns of postnatal growth, and compare them to intrauterine growth. By assessing the weight growth rate, we show three phases of postnatal growth not evident in intrauterine growth. Furthermore, postnatal length and head circumference growth exhibit a slow rate after birth, followed by an acceleration. Collectively, postnatal and intrauterine growth are distinctly different. Although postnatal growth models do not represent optimal growth of very preterm infants, they can serve as a practical tool for clinical assessment of growth and for nutrition research.

In 2021, 15% (57,096 out of 382,817) of preterm infants in the United States were born before 32 weeks' gestation[1]. Plotting weight, length, and head circumference on growth charts is integral to care for preterm infants, as assessment of growth is required for fluid and nutritional management and for identifying infants with abnormal growth[2].

In 1977, the American Academy of Pediatrics Committee on Nutrition suggested that "the goal of feeding regimens for low-birth-weight infants is to obtain a prompt postnatal resumption of growth to a rate approximating intrauterine growth because this is believed to provide the best possible conditions for subsequent normal development"[3]. In 1999, Ehrenkranz et al. investigated "longitudinal growth of hospitalized very low birth weight (VLBM) infants" with a goal "to better understand postnatal growth, to help identify infants developing illnesses affecting growth, and to aid in the design of future research"[4]. The longitudinal growth models were developed based on a 100-gram increment of birth weight between 501 and 1500 grams using a modeling approach that allowed for the capture of the nonlinear growth pattern. In 2006, by assessing a subset of the cohort with

birth weight between 501 and 1000 grams, Ehrenkranz et al. further showed that a poor rate of growth was associated with adverse neurodevelopmental and growth outcomes at 18 to 22 months' corrected age[5].

In 2013, Fenton and Kim revised the 2003 Fenton growth charts by conducting a meta-analysis of six published growth charts that included preterm infants[6–13]. The 2013 Fenton growth charts are based on cross-sectional anthropometric measurements obtained at birth at each specific gestational week from 22 through 36 weeks of gestation. Between 36 and 50 weeks (not including 36 and 50 weeks), only cubic splines were applied in order to allow the growth curves to be connected to the 2006 World Health Organization Child Growth Standards at 50 weeks[12,14]. The 2013 Fenton growth charts, thought to represent intrauterine growth, are commonly used to guide growth monitoring and nutritional management of very preterm infants worldwide.

Although direct comparisons between the Ehrenkranz postnatal growth charts and the 2013 Fenton intrauterine growth charts are not

[1]Department of Neonatology, Kaiser Permanente Riverside Medical Center, Riverside, CA, USA. [2]Clinician Investigator Program, Southern California Permanente Medical Group, Pasadena, CA, USA. [3]Division of Health Services and Outcomes Research, Children's Mercy Research Institute, Kansas City, MO, USA. [4]Department of Pediatrics, School of Medicine, University of Missouri-Kansas City, Kansas City, MO, USA. [5]Center for Research, Education, Quality and Safety, Pediatrix® Medical Group, Sunrise, FL, USA. ✉e-mail: Fu-Sheng.X.Chou@kp.org

possible, differences are noticeable upon visual inspection of the reference graphs from each publication. First, Ehrenkranz et al. showed postnatal growth began with weight loss, compared to no weight loss in intrauterine growth as shown in the Fenton charts; second, between 27 and 34 weeks postmenstrual age (PMA), the Fenton charts showed higher intrauterine weight growth rate among lower percentiles, whereas in the Ehrenkranz charts, preterm infants in the higher birth weight categories were shown to always grow at a higher rate (Table 3 of ref. 4); third, the intrauterine weight deceleration during the second half of the third trimester shown in the Fenton charts was not observed in postnatal growth; fourth, postnatal length and head circumference growth follow a concave pattern (Figs. 2 and 3 of ref. 4), compared to a convex intrauterine pattern as shown in the Fenton charts.

The Ehrenkranz study was published more than two decades ago. Since then, neonatal nutrition research has resulted in significant changes in clinical nutritional practice. The objectives of our study are to investigate whether postnatal growth based on measurements recorded more recently have achieved a rate approximating the intrauterine growth rate, and to evaluate whether a common pattern of postnatal growth exists among very preterm infants of various gestational age (GA) groups, sexes, and sizes at birth.

Here, we show that the postnatal growth of very preterm infants differs from the 2013 Fenton intrauterine growth charts. The rate of postnatal weight, length, and head circumference growth do not approximate that of intrauterine growth. The intrauterine and postnatal growth patterns diverge from birth and remain distinctly different throughout the assessment period.

## Results

We obtained nearly 5.5 million total weight data points, over 750,000 total length data points, and over 1.3 million head circumference data points from 89,218 infants born between 2010 through 2020 for the study (Table 1). The number of measurement values and the associated demographic information stratified by sex are available in Table 1. The above information further stratified by the GA groups are available in Table S1. Morbidity and mortality data are available in Table S2.

### Table 1 | Demographic characteristics

|  | Total | Female | Male |
|---|---|---|---|
| **Number of infants** | 89,218 | 41,798 | 47,420 |
| **Number of Data points** | | | |
| Weight | 5,478,055 | 2,602,008 | 2,876,047 |
| Length | 755,687 | 358,968 | 396,719 |
| Head Circumference | 1,316,087 | 620,079 | 696,008 |
| **Race/ethnicity** | | | |
| White | 36,628 (41%) | 16,930 (40%) | 19,698 (43%) |
| Black | 25,577 (29%) | 12,537 (30%) | 13,040 (27%) |
| Hispanic | 17,008 (19%) | 7772 (19%) | 9236 (19%) |
| Asian | 2643 (3%) | 1179 (3%) | 1464 (3%) |
| Other | 7362 (8%) | 3380 (8%) | 3982 (8%) |
| **Infant size at Birth** | | | |
| < 10th percentile | 8.1% | 8% | 8.3% |
| > 90th percentile | 6.4% | 6% | 6.7% |
| **Cesarean Delivery** | 63,161 (71%) | 30,042 (72%) | 33,119 (70%) |
| **Length of NICU Stay**[a] | 61 (40, 86) | 61 (42, 87) | 61 (39, 86) |
| **Discharge Status** | | | |
| Home | 67,694 (76%) | 32,391 (78%) | 35,303 (75%) |
| Transfer to another facility | 12,165 (14%) | 5444 (13%) | 6721 (14%) |
| Deceased | 9359 (10%) | 3963 (9%) | 5396 (11%) |

[a]Data presented are for all infants including those who were alive upon NICU discharge and those who died before NICU discharge.

We inspected postnatal growth by plotting measurement values for weight, length, and head circumference on the 2013 Fenton intrauterine growth charts (Fig. 1 top row of each sex group shows representative plots of measurement values from infants born at 25 weeks 0 days together with the 2013 Fenton growth curves; plots of all GA and sex groups are available in Figures S1-S8). 70–90% of the measurements fell below the 50th percentile lines, and 25–50% fell below the 10th percentile lines (Table S3).

We validated the postnatal growth models using metrics, including MAE, RMSE, and R-squared, and found no evidence of overfitting. Specifically, as shown in Figure S9 and S10, the metrics from the training dataset (randomly selected 70% of the infants) and the validation dataset (remaining 30% of the infants) were plotted closely to each other. Only up to 0.13% of the measurement values met the criteria for the outlier, so no values were excluded (Table S4). The postnatal growth models matched the actual measurement values better than the 2013 Fenton growth charts did, with the percentile lines and the percentages of the measurement values that fell below them more closely approximated with each other (Fig. 1 middle row of each sex group, Figures S1-S8, and Table S5).

The postnatal growth models followed a pattern similar to the Ehrenkranz postnatal charts[14] and were different from the Fenton charts[12] (Fig. 1 bottom row of each sex group and Figure S1-S8). The postnatal growth models captured initial weight loss, with the lowest weight point estimated to occur on the day of life (DOL) 4 (All models start on DOL0). The models also estimated birth weight to be regained between DOL8 and DOL10 (Table 2). Using the postnatal growth models to calculate birth weight percentiles, we found that, depending on the GA and sex groups, between 1.1 and 10.5% of the infants had birth weight above the 90th percentile, and between 0.5 and 12.4% of infants had birth weight below the 10th percentile.

Postnatal and intrauterine growth rates were plotted together for comparison (Fig. 2 shows plots of the modeled growth rates for the 25-week GA group together with the growth rates calculated from the Fenton growth charts; the growth rate plots of all birth GA groups are available in Figure S11-S18). Several differences were observed:

*Weight:* Our postnatal growth models captured the initial weight loss, followed by an acceleration that was slower than intrauterine weight growth acceleration (Fig. 2a, d and Figure S11-S18). Intrauterine weight growth rates reached their maximum at 33/34 weeks PMA in all percentile lines. Postnatal weight growth rates did not approximate the maximum intrauterine growth rates until after 34 weeks PMA, and the approximation only occurred with the 90th and 97th percentile lines. For the 3rd, 10th, and 50th percentile lines, the rates of weight gain never reached the maximum intrauterine growth rates. The maximum postnatal weight gain of the 50th percentile line approximated the maximum intrauterine growth rate of the 3rd percentile line (red horizontal lines in Fig. 2a, d and Figures S11-S18). Postnatal weight gain reached a steady state after reaching 34 weeks PMA without deceleration. In contrast, intrauterine weight decelerated between 33/34 weeks and 38/39 weeks PMA. The degree of intrauterine weight growth deceleration trended inversely with the percentile lines.

*Length:* Postnatal length growth began slower than intrauterine length growth but accelerated (Fig. 2b, e solid lines and Figure S11-S18), giving the growth pattern a concave shape (Fig. 1 and Figures S1-S8). Infants born with a length at the 90th percentile or higher had a greater acceleration of length growth than infants with a birth length at the 10th percentile or lower. However, the maximum postnatal growth rate for each percentile never reached the corresponding maximum intrauterine growth rate. The accelerations/decelerations seen on the intrauterine growth reference were distinctly different from postnatal length growth rate changes and were dependent on the percentiles: the intrauterine growth rates of higher percentiles were initially higher but

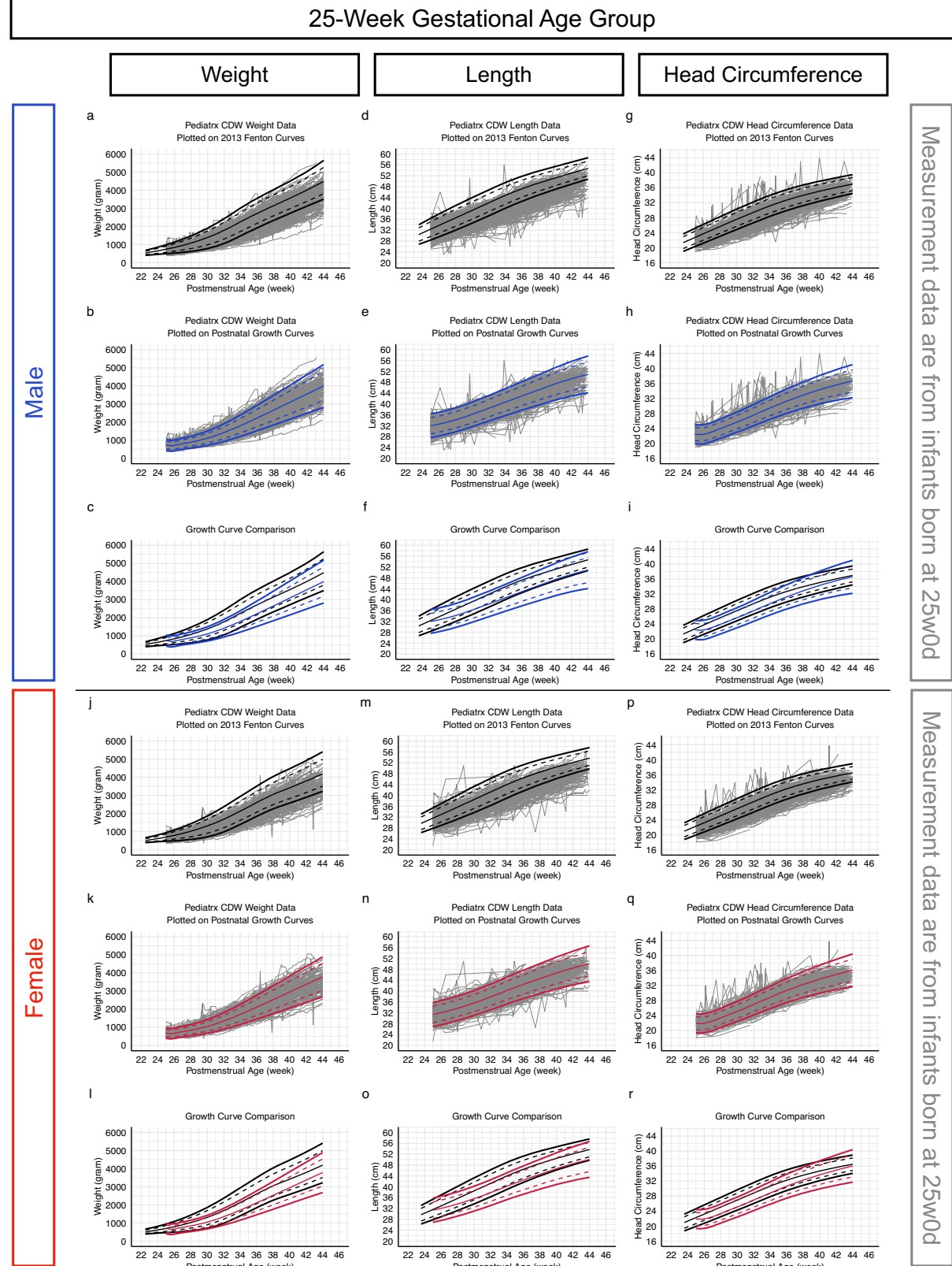

**Fig. 1 | Plotting measurement values on the 2013 Fenton growth charts and the postnatal growth models for comparison.** Representative plots of weight (left column; panels **a**, **b**, **c**, **j**, **k**, **l**), length (middle column; panels **d**, **e**, **f**, **m**, **n**, **o**), and head circumference (right column; panels **g**, **h**, **l**, **p**, **q**, **r**) measurement values from infants born at 25 weeks 0 days of gestation are overlaying on 2013 Fenton growth charts (top row of each sex group; panels **a**, **d**, **g**, **j**, **m**, **p**) and postnatal growth model for the 25-week gestation age group (middle row of each sex group; panels **b**, **e**, **h**, **k**, **n**, **q**). In the bottom row of each sex group (panels **e**, **f**, **i**, **l**, **o**, **r**), the 2013 Fenton growth charts (black) and the postnatal growth charts for the 25-week gestation age group (blue for male, red for female) are plotted together for comparison. Plots for all gestational age groups are available in Supplemental Fig. S1-S8. Source data are provided in the Source Data file.

**Table 2 | Characterization of model-derived estimates at birth (day of life 0) and during the initial weight loss phase**

| Gestational Age Group<br>Birth Gestational Age Range | 23<br>22w4d-<br>23w3d | 24<br>23w4d-<br>24w3d | 25<br>24w4d-<br>25w3d | 26<br>25w4d-<br>26w3d | 27<br>26w4d-<br>27w3d | 28<br>27w4d-<br>28w3d | 29<br>28w4d-<br>29w3d | 30<br>29w4d-<br>30w3d |
|---|---|---|---|---|---|---|---|---|
| **Female Infants** | | | | | | | | |
| **Birth Weight Distribution on the Postnatal Weight Models**[a] | | | | | | | | |
| *<10 th percentile* | 0.5% | 3.3% | 7.7% | 10.7% | 11.8% | 12.3% | 12.4% | 12.0% |
| *> 90 th percentile* | 1.5% | 2.0% | 3.8% | 7.0% | 7.8% | 9.6% | 10.2% | 10.5% |
| **DOL of Estimated Weight Nadir**[b] (day) | 4 | 4 | 4 | 4 | 4 | 4 | 4 | 4 |
| **Percent Loss between Estimated Birth Weight and Nadir**[b] (%) | 5.0 | 5.7 | 7.0 | 7.3 | 7.8 | 7.9 | 7.8 | 7.4 |
| **DOL when Estimated Birth Weight Regained**[b] (day) | 9 | 9 | 9 | 10 | 10 | 10 | 10 | 10 |
| **Male Infants** | | | | | | | | |
| **Birth Weight Distribution on the Postnatal Weight Models**[a] | | | | | | | | |
| *< 10 th percentile* | 1.0% | 2.5% | 6.9% | 10.2% | 11.3% | 11.9% | 11.6% | 12.1% |
| *> 90 th percentile* | 1.1% | 2.7% | 3.9% | 5.8% | 8.2% | 8.8% | 8.8% | 9.9% |
| **DOL of Estimated Weight Nadir**[b] (day) | 4 | 4 | 4 | 4 | 4 | 4 | 4 | 4 |
| **Percent Loss between Estimated Birth Weight and Nadir**[b] (%) | 4.9 | 5.8 | 6.8 | 6.9 | 7.7 | 7.6 | 7.4 | 6.9 |
| **DOL when Estimated Birth Weight Regained**[b] (day) | 8 | 9 | 9 | 9 | 10 | 10 | 10 | 10 |

[a]The percentage numbers were obtained by first calculating the corresponding percentiles of the birth weight measurement values using the postnatal growth models as the reference tool, followed by calculating the percentage of the percentiles that are less than 10 or greater than 90.
[b]The values used to calculate percent loss were derived from the estimates of the postnatal growth models, rather than the raw measurement values of the infants included in the study.
*DOL* day of life, *BW* birth weight.

**Table 3 | Model-derived average weight growth rates according to the percentiles during the weight acceleration and the stable weight gain phases**

| Percentile | 3rd | 10th | 25th | 50th | 75th | 90th | 97th |
|---|---|---|---|---|---|---|---|
| **Female** | | | | | | | |
| Growth rate during the weight acceleration phase (birth weight regained to 34 weeks PMA), gram/kg/day | 18.7±1.8 | 17.9±1.4 | 17.2±1.3 | 16.7±1.2 | 16.3±1.2 | 16.0±1.3 | 15.8±1.3 |
| Growth rate during the stable weight gain phase (34-44 weeks PMA), gram/day | 21.7±1.4 | 24.3±1.4 | 26.9±1.3 | 29.8±1.3 | 32.8±1.3 | 35.4±1.4 | 38.0±1.4 |
| **Male** | | | | | | | |
| Growth rate during the weight acceleration phase (birth weight regained to 34 weeks PMA), gram/kg/day | 18.5±1.7 | 17.6±1.4 | 17.0±1.2 | 16.5±1.2 | 16.1±1.2 | 15.9±1.2 | 15.6±1.2 |
| Growth rate during the stable weight gain phase (34-44 weeks PMA), gram/day | 22.7±1.4 | 25.5±1.3 | 28.2±1.3 | 31.3±1.4 | 34.4±1.4 | 37.2±1.5 | 40.0±1.6 |

Data presented as mean ± sd.
*PMA* postmenstrual age.

became lower after 32/33 weeks PMA due to greater deceleration in growth velocity (Fig. 2b, e dashed lines and Figures S11-S18).
*Head circumference:* During the first 2 weeks after birth, there was minimal head circumference growth, which was followed by growth acceleration surpassing expected intrauterine growth (Fig. 2c, f and Figures S11-S18). The differences between postnatal and intrauterine growth rates were more pronounced in infants born with a head circumference at the 90th percentile or higher than infants born with a head circumference at the 10th percentile or lower.

Our study evaluated weight growth and growth rates to discern a common pattern that is followed by all GA groups, sexes and sizes at birth (Fig. 3 shows estimated postnatal weight growth as well as the growth rates in gram/kg/day and gram/day for the 50th percentile; results for the 3rd, 10th, 25th, 50th, 75th, 90th and 97th percentiles are available in Figure S19-S25).

Our models captured initial weight loss (Phase I), resulting in our charts' being different from the 2013 Fenton growth reference charts from the beginning (Fig. 3 top row and Figures S19-S25). The estimated percentages of weight loss and the DOL when the estimated

birth weight was regained are summarized in Table 2. The corresponding negative growth rates in gram/day and gram/kg/day are shown in Fig. 3.

Following the initial weight loss, weight gain entered an acceleration phase (Phase II), with a growth rate being roughly a function of current weight, until 34 weeks PMA, giving a near-horizontal appearance of the growth velocity curves when presented in gram/kg/day (Fig. 3 middle row and Figure S19-S25). Weight gain velocity in gram/kg/day in this phase was higher at the lower percentiles and lower at the higher percentiles, and was similar in both sexes for each corresponding percentile (Table 3). After 34 weeks PMA, weight gain entered a stable phase (Phase III) and became independent of current weight, giving a near-horizontal appearance of the growth velocity curves when presented in gram/day (Fig. 3 bottom row and Figure S19-S25). During this phase, higher percentiles had higher average weight gain (in gram/day) than lower percentiles (Table 3). Moreover, weight gain was higher in male than in female infants (Table 3). Throughout the modeled period, estimated weight, length, and head circumference were higher in male than in female infants (Figure S26).

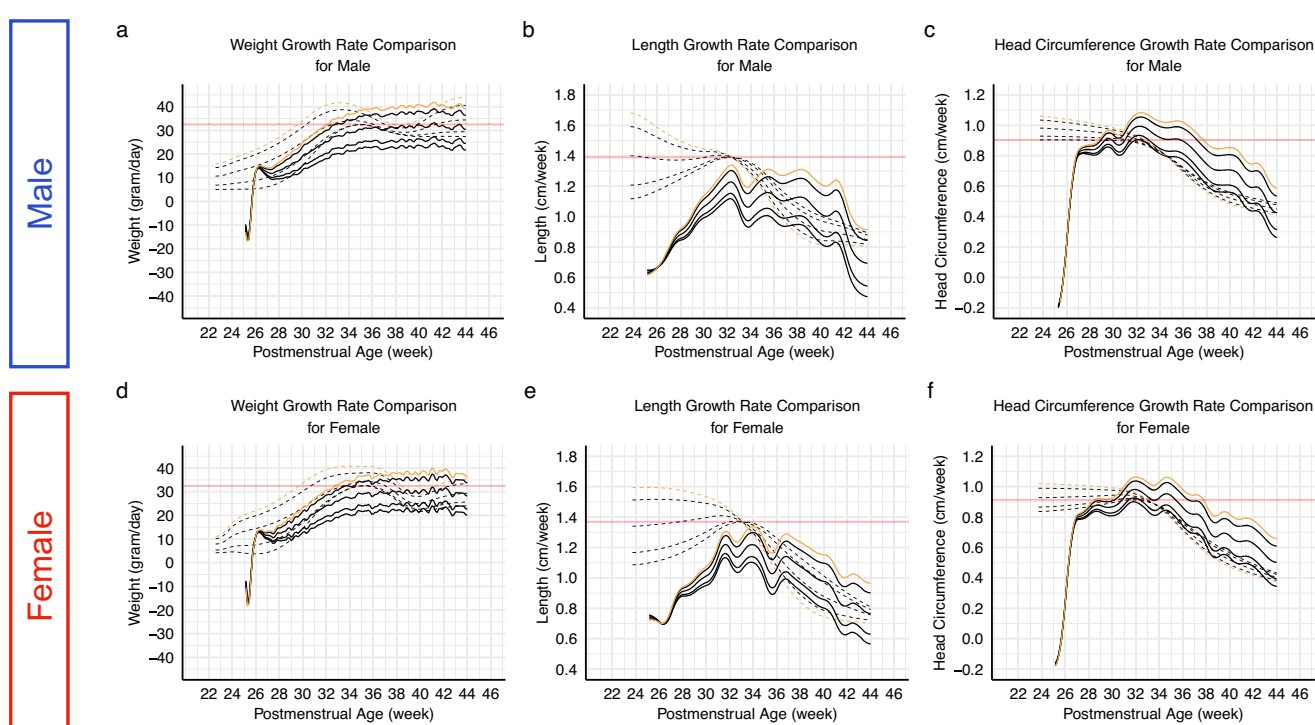

**Fig. 2 | Comparing postnatal and intrauterine growth rates.** Representative plots of the growth rates of weight (left column; panels **a**, **d**), length (middle column; panels **b**, **e**), and head circumference (right column; panels **c**, **f**) from the 25-week gestational age postnatal growth model (solid lines) and from the 2013 Fenton growth charts (dashed lines) are plotted together for comparison. Yellow lines represent the growth rates of the 97th percentile lines and the black lines represent the 90th, 50th, 10th, and 3rd percentile lines in the order of their distances from the yellow lines. Red horizontal lines denote the maximum intrauterine growth rates of the 3rd percentile lines from the 2013 Fenton charts. Plots for all gestational age groups are available in Supplemental Figure S11-S18. Source data are provided in the Source Data file.

To facilitate readers' access to the postnatal growth models, we have developed a web application (WebApp). To visit, please go to https://nicugrowth.app.

## Discussion

We used a large-scale real-world anthropometric dataset from 2010 through 2020 to assess the postnatal growth of very preterm infants. Similar to the approach used by INTERGROWTH-21st Preterm Postnatal Growth Standards[14], we modeled postnatal growth longitudinally by considering repeated measurements of individual infants. Different from INTERGROWTH-21st, our models allowed relaxation of parametric assumptions and did not purposely only include healthy and non-intrauterine-growth-restricted infants[15]. Our findings suggest that postnatal growth patterns of very preterm infants based on measurements recorded between 2010 through 2020 are similar to the growth pattern reported by Ehrenkranz et al. in 1999, and are different from the 2013 Fenton growth charts. Despite decades of work directed at monitoring and optimizing the growth of preterm infants[16–22], postnatal weight, length, and head circumference growth do not approximate intrauterine growth. The intrauterine and postnatal growth patterns diverge from birth and remain different throughout the assessment period. While postnatal growth failure in very preterm infants continues to be documented in the literature as a significant healthcare issue, our findings led to our reservations about the adequacy of the growth reference guide used in those studies[23–26].

A common growth pattern for all GA groups, sexes, and sizes at birth suggests that growth of very preterm infants after birth may be reprogrammed so they could adopt an extrauterine growth pattern under higher oxygen tension and without physical constraint. Although Ehrenkranz et al.[16]. did not examine the different phases of growth, the three intervals at the patient level that they included in their models to capture the nonlinear pattern of growth coincided with the three phases of growth in our findings. Ehrenkranz's first two intervals were based on quadratic terms of time, indicating growth accelerations/decelerations, and therefore are consistent with the weight loss and the weight acceleration phases we described here. The linear term of time for the third interval indicated a constant weight gain velocity, which coincided with the stable weight gain phase we described here.

With our single-interval approach to longitudinal modeling at both infant and population levels, the percentile lines could be assembled without gaps. The percentile lines provide us an opportunity to compare differences in the rate of growth among infants with different sizes at birth and are essential components of clinically usable growth reference charts[23].

The current study has several limitations, including its retrospective nature, the lack of nutritional data for adjusting, and the omission of accounting for heterogeneity at the NICU level in the models. The CDW did not provide information on the measurement method in each NICU, including whether and how the weight of instruments such as respiratory apparatus was subtracted from the measured weight, whether an infant length board was used for length measurement, and what type of tape was used for head circumference measurement. Although it is typical practice to collect weight values daily, and length and head circumference weekly, the

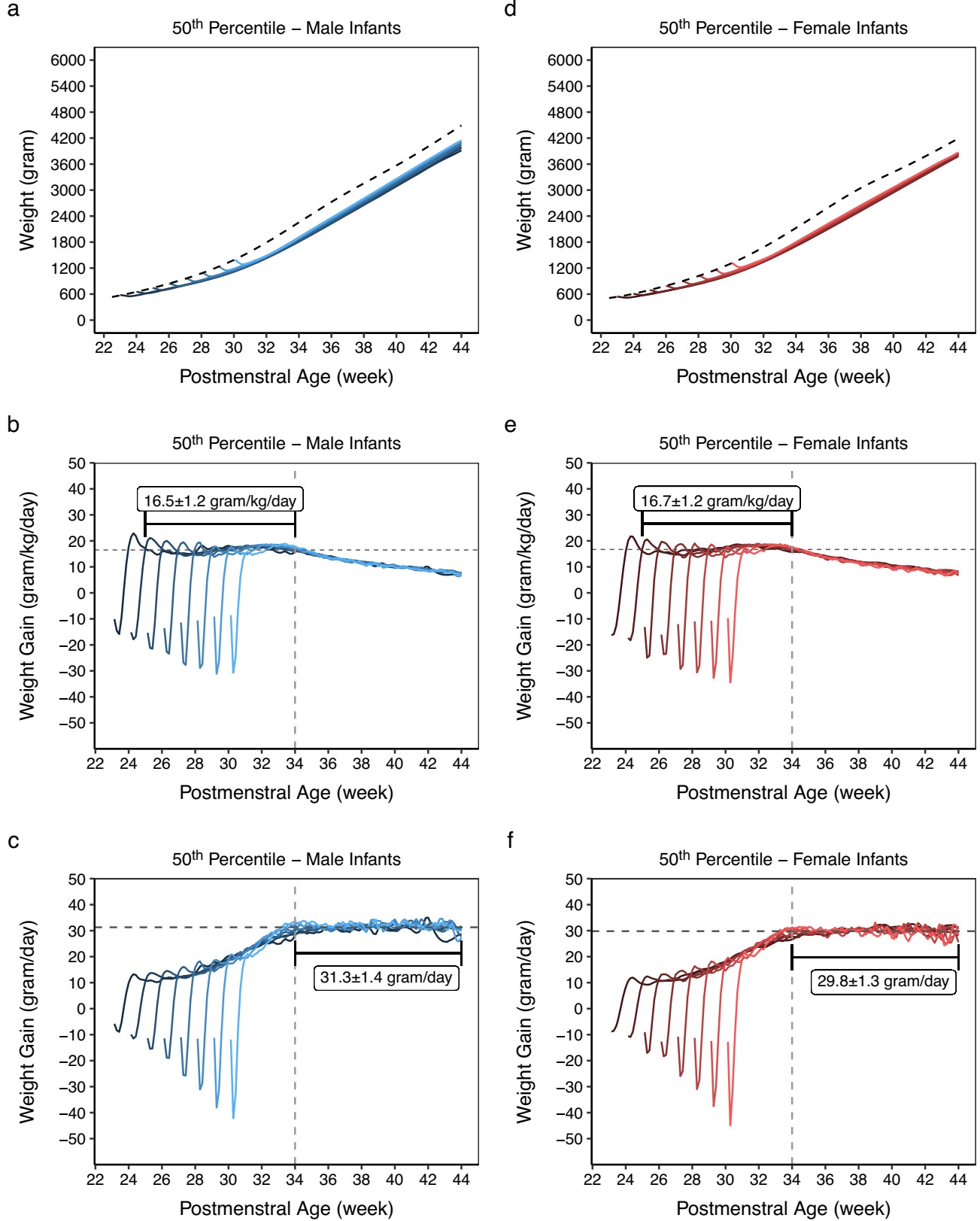

frequency may vary depending on the clinical condition of each infant and the NICU's policy. The estimates between 40 and 44 weeks PMA were based on a limited number of measurement, and the transition from Phase II to III may be attributed to less stringent nutrition supervision after infants reached the 34-week PMA milestone rather than physiological programming. Prospective growth

measurement controlling for nutrition provision would provide a more definitive answer.

In conclusion, we reaffirmed distinct patterns between postnatal and intrauterine growth among very preterm infants and identified three distinct phases of postnatal weight growth completely driven by the data. While these models may not represent the "optimal" postnatal

**Fig. 3 | The growth rates of the weight acceleration and the stable weight gain phases.** In the top row (panels **a**, **d**), weight growth estimates at the 50[th] percentile for all gestational age groups between 23 and 30 weeks are plotted together for comparison. The dashed line represents the 50[th] percentile line of the 2013 Fenton weight reference chart. In the middle row (panels **b**, **e**), estimates of weight growth rates in gram/kg/day for the 50[th] percentile of all gestational age groups between 23 and 30 weeks are plotted together for comparison. The horizontal dashed line denotes average weight gain of 16.5 gram/kg/day and 16.7 gram/kg/day for male and female, respectively, during the weight acceleration phase. In the bottom row (panels **c**, **f**), estimates of weight growth rates in gram/day for the 50[th] percentile of

all gestational age groups between 23 and 30 weeks are plotted together for comparison. The horizontal dashed line denotes average weight gain of 31.3 gram/day and 29.8 gram/day for male and female, respectively, in the stable weight gain phase. Numbers shown are mean ± sd of daily weight gain from all gestational age groups combined. The vertical line indicates 34 weeks postmenstrual age. For all panels, the color codes from dark to light denote the gestational age groups from 23 to 30 weeks. The blue gradients are for male, and the red gradients are for female infants. Plots for the 3[rd], 10[th], 25[th], 50[th], 75[th], 90[th], and 97[th] percentiles are available in Supplemental Figure S19-S25. Source data are provided in the Source Data file.

growth of very preterm infants, they can serve as a practical tool for clinical assessment of postnatal growth and for nutrition research.

## Methods

### Data source

Data were obtained from the Pediatrix® Clinical Data Warehouse (CDW), a large, multicenter, deidentified dataset that has been used in neonatal-perinatal research[27]. The CDW is generated by a proprietary standardized documentation and billing software tool, BabySteps®, created by Pediatrix® and used by participating NICUs in 35 states and Puerto Rico. The CDW contains deidentified clinical information representing approximately 20% of the NICU infants in the United States. The study was approved by the Research Advisory Committee of the Pediatrix® Center for Research, Education, Quality and Safety, and was exempted from the informed consent requirement by the Kaiser Permanente Southern California Institutional Review Board.

### Study design and data collection

This is an observational study using a retrospective real-world dataset. No statistical method was used to predetermine sample size. No randomization or blinding was required for this study. All infants with anthropometric records available between 2010 through 2020, regardless of being alive or dead, with a gestational age between 22 weeks 4 days and 30 weeks 3 days, a sex assigned typically based on physical features or genetic testing reports, and an admission age before DOL7 were included. Infants without sex assignment or without any measurement values available in the CDW were excluded. The presence or absence of antenatal exposure, congenital anomalies, multiple births, postnatal morbidities, or mortality were not used to determine infant inclusion or exclusion. Stillborn infants and infants who die in the delivery room are not admitted to the NICU and so are naturally excluded from the CDW.

All included infants had birth measurement data available, including birth weight, length, and head circumference. The electronic documentation tool does not have limits on birth measurement values. All variables listed in Table 1, Table S1, and Table S2 were extracted from the CDW. Procedures used to obtain anthropometric measurements were not available. Measurement values stored in the CDW were reported down to one gram for weight, and one millimeter for length and head circumference. Weight measurements were generally reported daily; length and head circumference measurements were generally reported weekly[28].

### Handling of missing values and outliers

Missing data exist in this real-world dataset because infants do not always have their anthropometric measurements documented at the same interval for the same duration. Missing data could be intermittent (e.g., measurement values were not entered into the medical charts or technical issues with the scales) or monotone (e.g., no data prior to the day of hospitalization at a Pediatrix® facility, and no data available after death, home discharge, or transferring outside Pediatrix® facilities). We argued that the reasons for these missing values were likely related to the infants' clinical condition rather than their growth data per se, and therefore assumed that they were all missing at

random (MAR). As a likelihood-based method, the generalized additive mixed modeling (GAMM) technique provides unbiased estimates under MAR using all available measurement values[29]. Therefore, there was no need for imputation.

Outliers were defined as having a leverage greater than three times the mean leverage and an absolute standardized residual greater than two. Outliers could only be assessed after models were developed. Notably, no data were excluded from the analyses before and after the outlier assessment.

### Postnatal growth modeling technique and model validation

We used GAMM to model postnatal growth patterns because GAMM does not require linear assumptions of the longitudinal data, allows the random effects to account for autocorrelations of repeated measurements for each infant, employ a maximum-likelihood approach to provide unbiased estimates (fitted values)[30]. GAMM has been used for modeling of anthropometric growth[31].

In this work, infants with the same sex and the same nearest complete week of gestation were grouped together for modeling. As an example, male infants born between 24 weeks 4 days and 25 weeks 3 days were grouped together in the 25-week GA male group. A single smooth function for DOL was used as the fixed effect. Thin plate regression splines were used for the smoothing panelty[32]. Measurement values from birth to a DOL corresponding to 44 weeks PMA were used as the outcome measure. Random intercept and random slope were included in the model. No growth phase was pre-specified during model development.

Model formula can be expressed as follows:

$$y_{ij} = f(t_{ij}) + u_0 + u_1 t_{ij} + \epsilon_{ij}, \, i = 1, \ldots, N; j = 1, \ldots, m_i \quad (1)$$

where

$y_{ij}$ is the outcome measure for the $j^{th}$ measurement of infant $i$,
$t_{ij}$ is DOL for the $j^{th}$ measurement of infant $i$,
$f(t_{ij})$ is the fixed-effect smooth function for $t_{ij}$,
$u_0$ and $u_1$ are random intercept and random slope, respectively, and
$\epsilon_{ij}$ indicates random/measurement error.

The $u_1 t_{ij}$ term is to capture the temporal dependency of repeated measurements at the infant level; it does not necessarily affect the estimates at the population level.

We first used measurement values from 70% of randomly selected infants in each GA and sex group to develop models and values from the remaining 30% of the infants for validation. Metrics used for model validation included mean absolute errors (MAE), root-mean-square errors (RMSE), and R-squared.

Estimating growth patterns of infants with different sizes at birth (the percentiles) requires the estimated standard deviation (SD), which can be calculated by taking the square root of the total variance ($Var(y)$). $Var(y)$ is calculated from the variances of (1) the random intercept ($u_0$), (2) the random slope ($u_1$), (3) the covariance between random intercept and random slope [$Cov(u_0, u_1)$], and (4) the error

term ($\epsilon$), using the following formula:

$$
\begin{aligned}
Var(y) \\
= Var(f(t) + u_0 + u_1 \times t + \epsilon) \\
= Var(f(t)) + Var(u_0) + t^2 \times Var(u_1) + 2 \times t \times Cov(u_0, u_1) + Var(\epsilon) \quad (2) \\
= 0 + Var(u_0) + t^2 \times Var(u_1) + 2 \times t \times Cov(u_0, u_1) + Var(\epsilon) \\
= Var(u_0) + t^2 \times Var(u_1) + 2 \times t \times Cov(u_0, u_1) + Var(\epsilon)
\end{aligned}
$$

where

$Var(y)$ is total variance,
$Var(f(t))$ is the variance of the fixed-effect smooth function, which is 0,
$Var(u_0)$ is the variance of the random intercept,
$Var(u_1)$ is the variance of the random slope,
$Cov(u_0, u_1)$ is the covariance between the random intercept and the random slope,
$Var(\epsilon)$ is the variance of the error term, and
$t$ is day of life (DOL).

In a post-hoc analysis, we calculated growth rates from the model-derived postnatal growth estimates at the $3^{rd}$, $10^{th}$, $25^{th}$, $50^{th}$, $75^{th}$, $90^{th}$, and $97^{th}$ percentiles by taking the difference in estimates between two consecutive days. As a convention, daily differences in length and head circumference (cm/day) were multiplied by 7 to produce weekly growth rates (cm/week). Growth phase determination was based on visual inspection of the growth rate curves. To compare, the intrauterine growth rates based on the 2013 Fenton charts were also calculated using the same method.

### Computational platform
All analyses were performed in R 4.1.1 on an Amazon Web Services instance running Ubuntu 20.04 LTS. The *dplyr* (v1.0.10) package was used for data cleanup. The *gamm4* (v0.2–6) package was used for growth trajectory modeling. The *lme4* (v1.1-31) package was used to calculate the total variance. The *ggplot2* (v3.4.0) *and patchwork* (v1.1.2) packages were used for plotting. The *shiny* (v1.7.4) package was used to develop the WebApp. The comprehensive R codes are provided in the Supplementary Software File.

### Reporting summary
Further information on research design is available in the Nature Portfolio Reporting Summary linked to this article.

## Data availability
The postnatal growth curves of all gestational age groups are available publicly by accessing the WebApp (https://nicugrowth.app), without any limitation. Tabular data of the postnatal growth curves developed in the current study cannot be shared openly due to concerns over the data being used for profit-generating purposes. Access can be obtained by sending a request via E-Mail to the corresponding author of the study (Dr. Fu-Sheng Chou, E-Mail: Fu-Sheng.X.Chou@kp.org), who will respond to the request within 7 days. A contract stating exclusive non-for-profit use by the requester, along with a signature, will be required for growth curve tabular data sharing. After receiving the document, Dr. Chou will share the data within 7 days. The authors do not have the permission to share the raw data supporting the current study due to the restriction of their contract with Pediatrix® Medical Group Center for Research, Education, Quality, and Safety. Raw data can be requested by contacting either Dr. Fu-Sheng Chou (Fu-Sheng.X.Chou@kp.org) or Dr. Reese H. Clark (reese.clark@pediatrix.com), who will respond to the request within 7 days. Once the request is approved by the Pediatrix® Medical Group Center for Research, Education, Quality, and Safety, Dr. Chou will share the data

within 7 days. The authors received growth charts data pertaining to the 2013 Fenton growth charts from Dr. Tanis R. Fenton from the University of Calgary but were not given permission to publicly share these data. Access to these data can be addressed to Dr. Tanis R. Fenton (tfenton@ucalgary.ca) who will review the request for research or individual use only and share the data within a reasonable time frame (in our experience it was within 7 days). Further details may be obtained by contacting Dr. Fenton directly. The raw and summarized data generated in this study and presented in the main figures are provided in the Supplementary Information/Source Data file. Source data are provided with this paper.

## Code availability
R codes for postnatal growth modeling, total variance calculation, and percentile value calculation are available in the Supplementary Software File. R codes for plotting and for the WebApp are available on GitHub (https://github.com/fschou-pxt/GTC-Website-Apps.git).

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

## Acknowledgements
The authors would like to thank Dr. Tanis R. Fenton, Ph.D. R.D. of the University of Calgary for sharing the LMS table for the 2013 Fenton growth charts for plotting use in this work. The authors would also like to thank Dr. Irene E. Olsen, Ph.D. R.D. L.D.N. of Drexel University and Dr. Veeral N. Tolia, M.D. of Pediatrix® Medical Group Inc. for critically reviewing an earlier version of the manuscript and their insightful comments.

## Author contributions
F.-S.C. led the study design and the analysis, and drafted the manuscript. H.-W.Y. supervised study design and the analysis, and drafted the manuscript. R.H.C. supervised study design, supplied measurement data for the analysis from the Pediatrix® Clinical Data Warehouse, and drafted the manuscript. All authors contributed to the interpretation of the results and critically reviewed and redrafted the manuscript.

## Competing interests
The authors declare no competing interests.
