## [Peer Review File · Nature Communications]

REVIEWER COMMENTS

Reviewer #1 (Remarks to the Author):

This is an excellent paper on postnatal growth patterns among very preterm infants. These data are important for clinicians and researchers. I have some relatively minor suggestions for the authors.

Title: Use of declarative title and some jargon such as "contemporary" (quickly becomes outdated) "real-world" and "large scale". I would recommend a non-declarative title but defer to the instructions for authors for this journal.

Abstract: "began slower" - consider rephrasing. "little to no" - minimal? Conclusion could be improved overall. "real-world" sounds jargony - remove? "proper" - hyperbole - remove? models "may" not represent optimal growth. How do you know if data is biased or not as do not know the exact method of collection, which might have introduced bias. "realistic" - hyperbole?

Introduction - well written.

Methods - "Advisor" or Advisory? The patient inclusion section should be expanded - What was the gestational age for inclusion, any weight limits for a given GA, did you include infants with a known birth weight and GA only, did you include infants with major anomalies etc. Could you discuss whether there are recommended techniques and frequency (daily v weekly) to obtain anthropometric measurements in the network, might also include whether to the nearest 10 grams or 1 gram, mm or cm etc.

Were the analyses by growth phase prespecified? - if not should state these were post-hoc and describe in methods.

Results: Was expecting a comment on model performance metrics from the split validation described in methods.

Limitations: CPAP hats or other measurement errors not accounted for, measured to mm versus cm, increments of grams etc., method of collection not standardized but this may be a non-differential bias in the study as each infant measured in the same non-ideal way, but also potentially a source of bias if technique changed over time and differed between units in the network. Did not account for illness severity in models but this may also be a potential strength compared with other datasets. Any reason why you did not include z-scores in this study?

Any comment on 5%-8% nadir (was this mean?) as less than typically reported in very preterm and extremely preterm infants.

Conclusion better 1st and 3rd sentence than in abstract. 2nd sentence is speculative and may not be necessary in conclusion.

Tables: missing some units of measure e.g. Birth weight (grams), mean+/-SD

Missing measures of variability for some data such as DOL and % weight loss also.

Confused by gestational age group versus birth gestational age range - this is missing from methods - why are these different? Did you include infants with missing birth weight data?

Reviewer #2 (Remarks to the Author):

Congratulations on the development of your well constructed and clinically important postnatal growth models for preterm infants.

Noteworthy results:

The authors have used a large dataset of longitudinal measurements to develop postnatal growth models, individualised for gestational age at delivery, fetal sex and birthweight centile. These show a three-phase growth pattern that I'm sure many clinicians will recognise from practice but which is not shown by the previous Fenton charts.

Significance:

These charts provide a significant methodological improvement from previous charts and give up-to-date data on observed growth patterns. This is important for patient counselling, clinical care and studying alternative nutritional approaches.

Critique of design, methods and conclusions:

The authors have used a large available data set with appropriate generalised mixed models. They describe the model assumptions, handling of missing data and justification for this and provide supplementary data on model characteristics and outputs. Their methods are largely reproducible (see below for suggested additional information).

The authors are appropriately circumspect with their conclusions, clearly stating that their models reflect the postnatal growth patterns we see in practice, not necessarily optimum postnatal growth.

Suggested changes:

1) Data source, line 103. It would be helpful if you had any data on whether / how the CDW patients represent the general patient population.

2) Line 108. Did you include multiple births? If so did you consider exploring whether there were any significant differences between curves generated on singletons vs multiples? In utero we see differences in expected growth between singletons and multiples (although it is still debated whether the smaller size in multiples is 'normal'). If you did include multiples it would be useful to give percentages.

3) Line 109. From line 147 it sounds as though you included those discharged alive from NNU and those who died. If so it might help to be a little more explicit about that in line 109 as when we talk about discharge in general conversation it usually implies alive.

4) Line 137. Which chart did you use to define birthweight centiles? Did you generate your own centiles based on birthweight data or did you use a reference chart?

Based on the fact that only 5-7% of your babies had a birthweight <10th centile at 29 and 30 weeks I wonder whether you used charts derived from birthweights (either yours or someone else). Usually we find birthweight charts give lower weight reference ranges than charts based on ultrasound estimated fetal weight because preterm babies are more likely to be small-for-gestational-age or growth restricted (given that overlapping risk factors for preterm birth and SGA). I would suggest using the WHO estimated fetal weight reference ranges to generate birthweight centiles (NOT the Intergrowth estimated fetal weight charts as they have poor fit below 30 weeks).

5) Results. You provide lovely figures comparing your findings with the Fenton charts but none providing a direct comparison with the Ehrenkranz curves. Is there a reason for this?

6) Have you thought about how you might make your findings easily useable for clinicians? I can imagine it would be very helpful to be able to generate an individualised 'expected growth trajectory' chart based on gestational age at delivery, sex and birthweight centile.

7) Discussion. It's refreshing to see a discussion that is so clear about the limitations of the study and what the findings do and do not mean. You could add a bit more about the benefits of your findings

though in terms of setting realistic expectations for clinicians and parents. Having a preterm baby on NNU is a rocky ride and the first few weeks especially can be difficult. I imagine being able to reassure parents that the drop in weight or limited weight gain initially is 'normal' would be very helpful for them.

The authors compare postnatal growth of very preterm infants to the 2013 Fenton intrauterine growth reference and evaluate the pattern of postnatal growth across gestational ages, sex, and size at birth. They have used Generalized Additive Mixed Models in order to estimate the growth curves. The study is interesting, however, with respect to the models, I consider that important details and clarity in their implementation are lacking.

- 1- Authors state in line 117 *GAMM does not require linear assumptions of the longitudinal data, contains the random effects to account for autocorrelations of repeated measurements for each infant, employ a maximum-likelihood approach to provide unbiased estimates (fitted values) when data is missing at random (MAR)*. This needs a reference. In addition, what is the amount of missing values? Did authors perform any statistical test to analyze if missing data could be considered as MAR or did they assume that?
- 2- Same sentence as in line 117 is repeated in line 150.
- 3- As authors say, GAMM do not require for linear assumptions of the longitudinal data. Therefore, some smooth function is considered to model the relationship between postmenstrual age and the outcome variables (Weight, Length, and Circumference). The authors should detail which function(s) (and estimation approaches) have been considered to estimate this relationship. Did they use cubic splines, p-splines, b-splines,...? Different approaches can be used. Also different estimation methods for the smooth functions can be used. See for example:

Carballo A, Durban M, Kauermann G, Lee D-J. A general framework for prediction in penalized regression. *Statistical Modelling*. 2021;21(4):293-312.
doi:10.1177/1471082X19896867

Eilers, P. H., Marx, B. D., & Durbán, M. (2015). Twenty years of P-splines. *SORT: statistics and operations research transactions*, 39(2), 0149-186.

Sánchez-González, M., Durbán, M., Lee, D. J., Cañellas, I., & Sixto, H. (2017). Smooth additive mixed models for predicting aboveground biomass. *Journal of Agricultural, Biological and Environmental Statistics*, 22, 23-41.

Wood, S. N. (2017). *Generalized additive models: an introduction with R*. CRC press.

- 4- Authors say in line 131 that variances Gaussian assumption is considered. That paragraph confuses me. The Gaussian assumption is for the response variable and the error of the model (see for instance the reference Wood, 2017 above). If authors want to specify the model, they should write the model expression rather than the variance. In addition, in the expression written in lines 133 and 134, a linear relationship between DOL and the response (y) is assumed. They should correct that. In addition, do response variables belong to a Gaussian distribution?
- 5- Results are given by GA and sex groups. Have these variables been introduced in the GAMM model as fixed effects?
- 6- Maybe I missed it, but I did not see what software or functions were used to estimate the curves.
- 7- A weir effect is shown in Figure 3(B) and Figure 3(C) between weeks 22 and 32. Could authors explain that? Maybe showing confidence intervals for the estimates could help.

REVIEWER COMMENTS

Reviewer #1 (Remarks to the Author):

This is an excellent paper on postnatal growth patterns among very preterm infants. These data are important for clinicians and researchers. I have some relatively minor suggestions for the authors.

Title: Use of declarative title and some jargon such as "contemporary" (quickly becomes outdated) "real-world" and "large scale". I would recommend a non-declarative title but defer to the instructions for authors for this journal.

We have revised the title to "A Comparative Study of Postnatal Anthropometric Growth in Very Preterm Infants and Intrauterine Growth" to avoid use of declarative title.

Throughout the main text, we also replaced contemporary with "measurement recorded between 2010 through 2020".

Abstract:

"began slower" - consider rephrasing. we rephrase to "exhibited a slower rate of onset"

"little to no" - minimal? We changed to "minimal"

Conclusion could be improved overall. We took the 1st and 3rd sentences of Conclusion from the main text

"real-world" sounds jargony - remove? removed

"proper" - hyperbole - remove? removed

models "may" not represent optimal growth. We changed "do" to "may"

How do you know if data is biased or not as do not know the exact method of collection, which might have introduced bias. "realistic" - hyperbole?

Thank you for pointing this out. "Unbiased" as compared to the Fenton charts when plotted together with the raw data, but we agree that it causes confusion. We have revised the conclusion section of the abstract using the conclusion in the main text.

Introduction - well written. Thank you.

Methods -

"Advisor" or Advisory?

Advisory. Thank you.

The patient inclusion section should be expanded -

What was the gestational age for inclusion, any weight limits for a given GA, did you include infants with a known birth weight and GA only, did you include infants with major anomalies etc. Could you discuss whether there are recommended techniques and frequency (daily v weekly) to obtain anthropometric measurements in the network, might also include whether to the nearest 10 grams or 1 gram, mm or cm etc.

Thank you for listing the missing information. The paragraph has been revised and expanded.

Regarding the frequency of measuring procedures, we have also revised the Limitations paragraph in Discussion to include this. Please refer to the updated version of the manuscript for more details.

Were the analyses by growth phase prespecified? - if not should state these were post-hoc and describe in methods.

We updated the Methods section in the subsection of "Modeling of postnatal growth" to emphasize that the growth phase was not pre-determined.

Results: Was expecting a comment on model performance metrics from the split validation described in methods.

We have revised the first sentence of the second paragraph in the "Postnatal vs. Intrauterine growth comparison" subsection under Results as follows for clarity:

"We validated the postnatal growth models using metrics including MAE, RMSE, and R-squared, and found no evidence of overfitting as shown in Figure S1 and S2, where the metrics from the training dataset (randomly selected 70% of the infants) and the validation dataset (remaining 30% of the infants) were closely aligned."

Limitations: CPAP hats or other measurement errors not accounted for, measured to mm versus cm, increments of grams etc., method of collection not standardized but this may be a non-differential bias in the study as each infant measured in the same non-ideal way, but also potentially a source of bias if technique changed over time and differed between units in the network. Did not account for illness severity in models but this may also be a potential strength compared with other datasets.

Thank you for pointing out these additional limitations. We have incorporated the information you provided into the Limitations paragraph.

Any reason why you did not include z-scores in this study?

We chose to present percentiles throughout the manuscript, as it is the convention in growth reference charts, even though Z-scores and percentiles are interchangeable under a Gaussian assumption. In the WebApp we have developed, both z-scores and percentiles are displayed.

Any comment on 5%-8% nadir (was this mean?) as less than typically reported in very preterm and extremely preterm infants.

We regret any confusion caused. The rows in Table 1 relating to the day of life (DOL) of nadir and the percentage of weight loss should not have been included in Table 1. The percentage of weight loss between estimated birth weight and the estimated weight nadir was calculated using the model's estimates (which correspond to the 50th percentile growth trajectory), rather than the raw measurement values of each infant. We have moved this data to a newly created Table S5 and added a footnote for clarification. Infants who did not have a weight nadir available (those who died or were transferred out before nadir, or those who arrived at a Pediatrix NICU after the weight nadir) would need to be excluded to calculate the cohort percentage of weight loss at nadir and the DOL of nadir, and this would not reflect the data used for model development. Therefore, we did not pursue these descriptive analyses.

Conclusion better 1st and 3rd sentence than in abstract. 2nd sentence is speculative and may not be necessary in conclusion.

Thank you. We removed the 2nd sentence and updated the conclusion of the abstract accordingly.

Tables: missing some units of measure e.g. Birth weight (grams), mean+/-SD

The table has been updated to add the units of measure.

Missing measures of variability for some data such as DOL and % weight loss also.

As described above, the data regarding the DOL of nadir and the percentage of weight loss were based on model estimates rather than statistical summaries of raw data. To calculate measures of variability, we would need to conduct bootstrapping, which is a computationally intensive process and may not be practical due to our sample size. Each model we presented here took between 5-13 hours to complete using a high-end computing server, and conducting 50-100 iterations of bootstrapping would not be feasible with current computing power capabilities.

Confused by gestational age group versus birth gestational age range - this is missing from methods - why are these different? Did you include infants with missing birth weight data?

We apologize for any confusion caused. When planning this project and based on previous work, we knew that growth was dependent on both sex and gestational age. Additionally, due to technical limitations in fitting millions of measurement values in a single model, we made the decision to group infants by nearest gestational age. Specifically, we included infants born between X-1 weeks 4 days and X weeks 3 days in the X-week group, where X ranges from 23 to 30. This is the reason for the gestational age group in Table 1. We have provided an example of what we mean by "nearest gestational age" in the second paragraph of the "Modeling of postnatal growth" subsection under Methods. Also, all infants in the dataset had all three birth measurements (weight, length, and head circumference) available. We have included this information in the updated version of the manuscript.

Reviewer #2 (Remarks to the Author):

Congratulations on the development of your well constructed and clinically important postnatal growth models for preterm infants.

Noteworthy results:

The authors have used a large dataset of longitudinal measurements to develop postnatal growth models, individualised for gestational age at delivery, fetal sex and birthweight centile. These show a three-phase growth pattern that I'm sure many clinicians will recognise from practice but which is not shown by the previous Fenton charts.

Significance:

These charts provide a significant methodological improvement from previous charts and give up-to-date data on observed growth patterns. This is important for patient counselling, clinical care and studying alternative nutritional approaches.

Critique of design, methods and conclusions:

The authors have used a large available data set with appropriate generalised mixed models. They describe the model assumptions, handling of missing data and justification for this and provide supplementary data on model characteristics and outputs. Their methods are largely reproducible (see below for suggested additional information).

The authors are appropriately circumspect with their conclusions, clearly stating that their models reflect the postnatal growth patterns we see in practice, not necessarily optimum postnatal growth.

Suggested changes:

1) Data source, line 103. It would be helpful if you had any data on whether / how the CDW patients represent the general patient population.

This information has been added to Table 1.

2) Line 108. Did you include multiple births? If so did you consider exploring whether there were any significant differences between curves generated on singletons vs multiples? In utero we see differences in expected growth between singletons and multiples (although it is still debated whether the smaller size in multiples is 'normal'). If you did include multiples it would be useful to give percentages.

We included all infants that met gestational age and sex assignment criteria. We did not exclude any infants based on their being part of multiple births. The postnatal growth models we developed in this project will enable us to conduct subgroup analyses comparing different antenatal and postnatal conditions, including single vs. multiple births. We have added the percentages of multiple births for each gestational age group in Table 1 of the revised manuscript.

3) Line 109. From line 147 it sounds as though you included those discharged alive from NNU and those who died. If so it might help to be a little more explicit about that in line 109 as when we talk about discharge in general conversation it usually implies alive.

Thank you. The sentence has been updated to reflect both live and death discharges.

4) Line 137. Which chart did you use to define birthweight centiles? Did you generate your own centiles based on birthweight data or did you use a reference chart?

All percentiles presented in the manuscript were calculated using the variance formula, which allowed us to calculate the standard deviation for each day of life (DOL) and construct the percentile growth trajectory lines. The birth weight percentile was calculated using the mean estimate and standard deviation for DOL 0, based on the models developed in this project. We apologize for the confusion caused by presenting this information in Table 1, as the percentage was calculated using the growth models rather than a descriptive statistical summary based on existing growth charts. To clarify, we have moved the data from Table 1 to a newly created Table S5 and updated the text in the revised manuscript.

Based on the fact that only 5-7% of your babies had a birthweight <10th centile at 29 and 30 weeks I wonder whether you used charts derived from birthweights (either yours or someone else). Usually we find birthweight charts give lower weight reference ranges than charts based on ultrasound estimated fetal weight because preterm babies are more likely to be small-for-gestational-age or growth restricted (given that overlapping risk factors for preterm birth and SGA). I would suggest using the WHO estimated fetal weight reference ranges to generate birthweight centiles (NOT the Intergrowth estimated fetal weight charts as they have poor fit below 30 weeks).

We hope the previous explanation clarifies that the birth weight centiles were obtained from the models we developed.

5) Results. You provide lovely figures comparing your findings with the Fenton charts but none providing a direct comparison with the Ehrenkranz curves. Is there a reason for this?

We understand that the Ehrenkranz charts were published in 1999, and we are not aware of publicly available values for digital plotting. Moreover, the Ehrenkranz charts were based solely on birth weight category and not on gestational age. Therefore, it may not be appropriate to make a direct comparison between our charts and the Ehrenkranz charts. Please refer to Figure 1 of Dr. Ehrenkranz's paper (Ehrenkranz et al. 1999) for further information.

6) Have you thought about how you might make your findings easily useable for clinicians? I can imagine it would be very helpful to be able to generate an individualised 'expected growth trajectory' chart based on gestational age at delivery, sex and birthweight centile.

Indeed, while not being the focus of our study, Dr. Chou has developed a WebApp with the aim of providing public access to the postnatal growth charts. For additional information, please visit nicugrowth.app. The URL has been added to the manuscript as the final sentence of the conclusion in the main text.

7) Discussion. It's refreshing to see a discussion that is so clear about the limitations of the study and what the findings do and do not mean. You could add a bit more about the benefits of your findings though in terms of setting realistic expectations for clinicians and parents. Having a preterm baby on NNU is a rocky ride and the first few weeks especially can be difficult. I imagine being able to reassure parents that the drop in weight or limited weight gain initially is 'normal' would be very helpful for them.

Thank you for your feedback. We have updated the last part of the Conclusion with the following rephrasing to imply that growth assessment using our postnatal growth models may allow care providers and families to set realistic expectations:

“While these models may not represent the “optimal” postnatal growth of very preterm infants, they can serve as a practical tool for clinical assessment of postnatal growth and for nutrition research.”

Reviewer 3

The authors compare postnatal growth of very preterm infants to the 2013 Fenton intrauterine growth reference and evaluate the pattern of postnatal growth across gestational ages, sex, and size at birth. They have used Generalized Additive Mixed Models in order to estimate the growth curves. The study is interesting, however, with respect to the models, I consider that important details and clarity in their implementation are lacking.

1. Authors state in line 117 *GAMM does not require linear assumptions of the longitudinal data, contains the random effects to account for autocorrelations of repeated measurements for each infant, employ a maximum-likelihood approach to provide unbiased estimates (fitted values) when data is missing at random (MAR)*. This needs a reference. In addition, what is the amount of missing values? Did authors perform any statistical test to analyze if missing data could be consider as MAR or did they assumed that?

Likelihood-based methods, such as GAMM, are known to provide unbiased estimates under the missing at random (MAR) assumption (Little and Rubin 2002). As pointed out by (van Buuren 2018) "... It is not possible to test MAR versus MNAR since the information that is needed for such a test is missing." As we could not perform a test to confirm that the missing data were MAR, we only assumed that the missing data were MAR by justifying that the reasons for missing were likely related to the infant's clinical condition rather than the growth data *per se*. We have added the reference of Little and Rubin 2002 to the revised manuscript.

2. Same sentence as in line 117 is repeated in line 150.

Removed the part where MAR was mentioned in line 117

3. As authors say, GAMM do not require for linear assumptions of the longitudinal data. Therefore, some smooth function is considered to model the relationship between postmenstrual age and the outcome variables (Weight, Length, and Circumference).

The authors should detail which function(s) (and estimation approaches) have considered to estimate this relationship. Did they use cubic splines, p-splines, bsplines,...?

Different approaches can be used. Also different estimation methods for the smooth functions can be used. See for example:

Carballo A, Durban M, Kauermann G, Lee D-J. A general framework for prediction in penalized regression. *Statistical Modelling*. 2021;21(4):293-312. doi:10.1177/1471082X19896867

Eilers, P. H., Marx, B. D., & Durbán, M. (2015). Twenty years of P-splines. *SORT: statistics and operations research transactions*, 39(2), 0149-186.

Sánchez-González, M., Durbán, M., Lee, D. J., Cañellas, I., & Sixto, H. (2017). Smooth additive mixed models for predicting aboveground biomass. *Journal of Agricultural, Biological and Environmental Statistics*, 22, 23-41.

Wood, S. N. (2017). *Generalized additive models: an introduction with R*. CRC press.

We used the thin plate regression splines method, the default method for the `gamm4()` function of the `gamm4` package, for the smooth function (Wood 2003). We added this information in the

first paragraph of the “Modeling of postnatal growth” subsection under the Method section with reference.

4. Authors say in line 131 that variances Gaussian assumption is considered. That paragraph confuses me. The Gaussian assumption is for the response variable and the error of the model (see for instance the reference Wood, 2017 above). If authors want to specify the model, they should write the model expression rather than the variance.

Thank you for bringing this to our attention, and we apologize for any confusion caused by our previous statement. GAMM assumes Gaussian distribution for the residuals at the infant level, not the variance. We meant to state so but somehow the sentence came out incorrectly. Since the information on Gaussian assumption is quite technical and may not be necessary in this manuscript, we removed the sentence in the revised manuscript.

In addition, in the expression written in lines 133 and 134, a linear relationship between DOL and the response (y) is assumed. They should correct that.

In addition, do response variables belong to a Gaussian distribution?

We kept the equation for calculating the total variance, which we expanded in order to be clear (line 133 and 134 of the original manuscript) and added the model expression as follows in the revised manuscript:

$$y_{ij} = f(t_{ij}) + u_0 + u_1 t_{ij} + \epsilon_{ij}, \quad i = 1, \dots, N; \quad j = 1, \dots, m_i, \text{ where}$$

y_{ij} is an outcome measure for the j^{th} measurement of infant i ,
 t_{ij} is DOL for the j^{th} measurement of infant i ,
 $f(t_{ij})$ is the fixed-effect smooth function for t_{ij} ,
 u_0 and u_1 are random intercept and slope, and
 ϵ_{ij} indicates random/measurement error.

The $u_1 t_{ij}$ term is to capture the temporal dependency of repeated measurements at the infant level; it does not affect the estimates at the population level.

5. Results are given by GA and sex groups. Have this variables been introduced in the GAMM model as fixed effects?

We did not include GA and sex groups as fixed effects in our modeling for the following reasons:

1. Currently available growth charts for preterm infants for clinical use are sex-specific, as it is well-established that female and male intrauterine growth follow different patterns. Given the large amount of data points we have, we decided to build separate models for male and female infants. For GA, we felt that it would make more sense to build models separately as the clinical course during the initial 2-3 weeks of the acute phase is different for infants born at lower GA compared to those born at a higher GA, as the morbidity risks are different.
2. Building separate models for each GA group would also avoid monotone missing data at the later days of life among infants born at a higher GA.

3. Each weight model we built for this project took 5-13 hrs on a high-end computer server on Amazon Web Services. Despite none of the authors having a background in computer science, it was deemed impractical to construct a single model using all data points with sex and gestational age added as fixed effects from the technical and computing resource standpoint.
6. Maybe I missed it, but I did not see what software or functions were used to estimate the curves.

Thank you for your helpful suggestion. In the revised manuscript, we have added a new subsection under the Method section titled "Computational platform," where we provide details on the computing resources used for this project.

7. A weir effect is shown in Figure 3(B) and Figure 3(C) between weeks 22 and 32. Could authors explain that? Maybe showing confidence intervals for the estimates could help.

The growth curves depicted in Figures 3(B) and 3(C) represent growth rates calculated by estimating the weight trajectory difference between two consecutive days, expressed in grams per day in Figure 3(B) and grams per kilogram per day in Figure 3(C). Each curve represents a specific gestational age group. The initial negative growth rate (the part where the curves point downward in the negative value range) corresponds to the weight loss that occurs during the initial weight loss phase, which was followed by a positive weight gain rate that reflects the subsequent increase in weight after the nadir.

We added "The corresponding negative growth rates in grams/day and grams/kg/day are shown in Figure 3B and 3C, respectively." to the paragraph where the initial weight loss phase was described (the 2nd paragraph of the subsection "three phases of the postnatal weight change" in Results) in the revised manuscript.

References:

1. Buuren, Stef van. 2018. *Flexible Imputation of Missing Data, Second Edition*. CRC Press.
2. Ehrenkranz, R. A., N. Younes, J. A. Lemons, A. A. Fanaroff, E. F. Donovan, L. L. Wright, V. Katsikiotis, et al. 1999. "Longitudinal Growth of Hospitalized Very Low Birth Weight Infants." *Pediatrics* 104 (2 Pt 1): 280–89.
3. "Fetal Growth Calculator." n.d. Accessed May 7, 2023. <https://srhr.org/fetalgrowthcalculator/#/>.
4. Little, Roderick J. A., and Donald B. Rubin. 2002. *Statistical Analysis with Missing Data*. Wiley.
5. Wood, Simon N. 2003. "Thin Plate Regression Splines." *Journal of the Royal Statistical Society. Series B, Statistical Methodology* 65 (1): 95–114.

REVIEWERS' COMMENTS

Reviewer #1 (Remarks to the Author):

My comments were all addressed and I have no further suggestions.

Reviewer #2 (Remarks to the Author):

Thank you for addressing all of our questions and suggestions.

Reviewer #3 (Remarks to the Author):

The authors have responded adequately to all comments and suggestions made.

My recommendation is to accept the paper. Nevertheless, I suggest that the authors make one last minor change:

On page 8, line 156, I would limit the paragraph to a simpler sentence such as for example:

"Estimating growth patterns of infants with different sizes at birth (the percentiles) requires the estimated standard deviation (SD) which was computed by means of the gamm4 package in R."

And I propose to remove the variance formula. The formula is not quite correctly developed and since it is not necessary for the understanding of the paper I would remove it.

Point-by-point responses to the reviewers' comments

Reviewer #1 (Remarks to the Author):

My comments were all addressed and I have no further suggestions.

Reviewer #2 (Remarks to the Author):

Thank you for addressing all of our questions and suggestions.

Reviewer #3 (Remarks to the Author):

The authors have responded adequately to all comments and suggestions made.

My recommendation is to accept the paper. Nevertheless, I suggest that the authors make one last minor change:

On page 8, line 156, I would limit the paragraph to a simpler sentence such as for example:

"Estimating growth patterns of infants with different sizes at birth (the percentiles) requires the estimated standard deviation (SD) which was computed by means of the gamm4 package in R."

We thank the reviewer for further pointing out the deficiency of the statement. We recognize that the summation of individual variances to obtain the total variance is not as simply stated, as coefficients for the variance of the random slope and the covariance between the random intercept and random slope need to be taken into consideration.

We updated the paragraph to the following:

Estimating growth trajectories for infants of different sizes (the percentiles) requires the estimated standard deviation (SD), which was calculated by taking the square root of the total variance (detailed in Supplemental Method). As the variance is a function of DOL, so is the SD, and thus may vary by DOL.

We also added the following to the *Computational platform* subsection:

The lme4 (v1.1-31) package was used to calculate the total variance.

And I propose to remove the variance formula. The formula is not quite correctly developed and since it is not necessary for the understanding of the paper I would remove it.

We thank the reviewer for further pointing out the deficiency of the formula. We recognized that the exponent "2" was missing in the coefficient for the random slope. As the total variance was calculated manually, we felt responsible for including the formula and the relevant R codes for total variance calculation in the manuscript. We moved the formula statement to Supplementary Information for interested readers. Please refer to the revised manuscript.